# OpenReview forum: "The Assistant Axis: Situating and Stabilizing the Default Persona of Language Models"
_ICML.cc/2026/Conference — ICML 2026 spotlight_

### Official Review · Reviewer_do8J · 2026-03-08

**Soundness:** 2
**Presentation:** 3
**Significance:** 3
**Originality:** 2
**Overall Recommendation:** 4
**Confidence:** 4

**Summary:**

This paper generalizes upon existing work on “persona vectors”, which abstract certain model traits as linear vectors in the representation space. It claims the discovery of the “assistant axis”, vectors across layers obtained through the difference-in-means of the default residual stream and the averaged varying persona residual streams. It finds that certain conversations can cause the model to drift away from the default, while intervening with the “assistant axis” can reduce persona-based jailbreak success rate and successfully recognize a user’s mental health emergency.

**Compliance With Llm Reviewing Policy:**

Affirmed.

**Final Justification:**

The authors addressed the concerns about contribution framing and unclear writing/lack of supporting evidence. The new experiments with additional adversaries yield positive signals, but also suffer from testing only 2 attacks. The overclaimed result has been clarified, but the 4 considered instances might not be sufficient to draw a more systematic conclusion. Thus, I maintain my evaluation of weak acceptance.

**Key Questions For Authors:**

See weaknesses. Minor: the paper contains many misrendered $<$ signs as $¡$. Please fix them.

**Limitations:**

See weaknesses.

**Strengths And Weaknesses:**

**Strengths.**

- A nice extension of the persona vectors series of work. The identification of the assistant axis shows promising practical potential to combat certain jailbreaks and harmful behaviors. The correlation between persona drift, projection on the assistant axis, and uncommon, unwanted behavior offers new insights into understanding LLMs.

- Extensive experiments across a large suite of personas were evaluated on a collection of benchmarks. Though a large fraction of data is synthetic, as acknowledged in the limitation section, the paper studies multiple open-source LLMs and obtains (mostly) consistent results, suggesting the robustness of the findings.

**Weaknesses.**

- **Contribution framing.** From the perspective of technical primitive, the work is a direct generalization of persona vectors, and the technique used is almost the same as that done in previous work. The supportive evidence of the **validity** of the "assistant axis", the PCA result, is somewhat buried in the main text. In my view, it would be beneficial for the authors to (1) directly acknowledge that the method to extract the axis is an adaptation of prior work, and (2) highlight the PCA and base v.s. instruct results in Appendix A.

- **The type of harmful behavior to be countered is limited.** For example, given the current literature of diverse jailbreak methodologies, the paper focuses on persona-based attacks. It is unclear whether other types of attacks (for example, white-box optimization-based, few-shot-based, and iterative-prompting-based attacks) also induce a similar phenomenon as the persona drift, and the intervention could effectively defend against them.

- **Inconsistent and overclaimed results.** For example, in Section 4.1, it is claimed that “in therapy-related conversations … or philosophical conversations …, models … ending up at much lower values than the other topics. This occurred for all three target models, with all three auditors”. However, the writing conversation induces a lower score than therapy for both gemma and llama, and thus effectively negates the main claim made in the main text. Moreover, only evaluating on this specific 4 categories is not sufficient to draw a scientific conclusion.

- **Missing explanations and insufficient supporting evidence.** For example, in Section 4.2, it states “the model’s position along the Assistant Axis depends most strongly on the most recent user message”. However, this conclusion cannot be drawn directly from the stated prediction result with embeddings. How is this result derived? Similarly, many results throughout this paper are illustrative and could be cherry-picked, such as the conclusion in Section 4.3, where only the Qwen model is tested. What would happen if other models were used under this test?

Overall, I believe the paper provides important and intriguing insights, but also suffers from issues such as overclaiming and potentially cherry-picked qualitative demonstrations.

---

> ### Author Rebuttal · Authors · 2026-03-30
>
> Thanks for your careful and specific critique! Several points exposed real gaps between our data and our writeup.
>
> **Contribution framing.** We agree and have made both changes (also raised by reviewer J34h): §3.1 now explicitly credits the diff-in-means lineage [Marks & Tegmark 2024; Arditi et al. 2024; Panickssery et al. 2024], and the PCA validation from Appendix A (>0.92 PC1 correlation across model pairs; base≈instruct PCs in Gemma) is pulled into §3 before the axis is defined, so the main body supports that the axis is reflecting something that is pre-existing within the models rather than arbitrarily crafted.
>
> **§4.1 therapy/philosophical drift overclaim.** The reviewer is correct and we overstated uniformity across domains. After revisiting Figures 19–21: philosophy drifts lowest in all 9 model/auditor combinations, coding is stable in all 9, but therapy and writing both tend to drift slightly depending on the combination. Generally, it appears that writing dips sharply early then partially recovers, while therapy drifts monotonically. We revised the text to reflect this.
>
> On only 4 domains: our chose domains span task-oriented (coding, writing) vs. interpersonal (therapy) vs. meta-reflective (philosophy). We don't mean to claim exhaustiveness of all LLM use-cases; the finding is that *some* domains drift and some don't, and the former is concerning because they involve at-risk users not deliberately attempting to jailbreak the model. We've added this framing to §4.1 and softened the wording.
>
> **§4.2 "the model’s position along the Assistant Axis depends most strongly on the most recent user message."** This is our fault as the supporting comparison was cut for space. We found that the embeddings of the previous turn could strongly predict where the ensuing response landed along the axis (R² 0.53-0.77, p < 0.001) but not the projection delta from its previous response (R²=0.10, p < 0.001), across the three models. That is, the model’s position along the Assistant Axis depends most strongly on the most recent user message rather than where it was before (though the user message itself is dependent on the context of the conversation). The deleted clause and R²=0.10 are restored in §4.2.
>
> **§4.3 Qwen-only illustration of persona drift being risky.** We have this for all three models. Correlations: Qwen r=-0.52, Gemma r=-0.44, Llama r=-0.39 and same fan shape of the scatter plot (near-zero harmful rate at high projections near the Assistant, rising harm rates at low projections away from the Assistant). The figure for the additional models has been added to Appendix E alongside a note in the main text.
>
> **Jailbreak scope.** We don't claim that our approach defends against all possible jailbreaks; persona-based jailbreaks are the natural test case for a paper about persona drift. Whether many-shot jailbreaks, etc. manifest as persona drift is an interesting open question (our prior is that activation capping would probably be effective if applied at every token) and we have noted this in §8.1.
>
> **Rendering.** Thanks for the note, we've fixed throughout :)

---

> > ### Author Rebuttal · Reviewer_do8J · 2026-04-01
> >
> > I thank the authors for the response. I believe the contribution framing and the two points in the missing explanations part will be addressed when the authors are allowed to update the draft. However, regarding the remaining concerns:
> >
> > - Jailbreak scope: it is ok to study persona-based jailbreaks since the paper is about persona drift; however, to get a clear picture of how and when the proposed methodology could and could not work, other jailbreak adversaries need to be considered. It is beneficial to know the limitations of the proposal for the general safety community.
> >
> > - Overclaimed result: if the revised text correctly reflects the result, then the conclusion in this part will be weakened since there is no consistent trend across models.
> >
> > Thus I choose to maintain my current score, which reflects a weak acceptance.

---

> > > ### Author Response · Authors · 2026-04-06
> > >
> > > Thanks for your continued engagement~ We ran a few additional steering + activation capping experiments to expand the jailbreak scope and address your concerns.
> > >
> > > **Re: §4.1 consistency.** We'd gently push back on "no consistent trend." The first-order pattern holds in all 9 model and auditor combinations: coding is most stable, philosophy is lowest, and therapy/writing both drift substantially below coding. What varies is only the *relative ordering* of therapy vs. writing, a second-order detail. The actionable claim (task-oriented domains stable; interpersonal/reflective domains drift) is consistent across every condition, and that is what the revised §4.1 states.
> > >
> > > **Re: jailbreak scope.** We agree this characterization is valuable and ran new evaluations during the discussion window on two attack families *not* covered in §5:
> > >
> > > - **In-the-wild jailbreak templates** [Shen et al., CCS 2024] applied to HarmBench behaviors [Mazeika et al., 2024] (n=954) — third-party attacks not all based on personas.
> > > - **Base64 encoding** [Wei et al., NeurIPS 2023] applied to the SORRY-Bench base instructions (n=440) — a *non-persona* obfuscation attack.
> > >
> > > ASR under the HarmBench classifier, baseline vs. our interventions:
> > >
> > > | Model | Attack | Baseline | Capping (p0.25, §5) | Additive (fixed coef.) |
> > > |---|---|---|---|---|
> > > | Qwen-3-32B | In-the-wild (HarmBench) | 13.6% | **8.6%** | 1.2% (−0.25) |
> > > | Qwen-3-32B | Base64 (SORRY-Bench) | 6.1% | **3.2%** | 0.7% (−2.0) |
> > > | Gemma-2-27B | In-the-wild (HarmBench) | 17.4% | n/a* | **2.2%** (−0.2) |
> > > | Gemma-2-27B | Base64 (SORRY-Bench) | 2.3% | n/a* | **0.2%** (−0.2) |
> > >
> > > *Gemma-2's pre/post-LN architecture precludes our capping implementation; we report additive steering along the same axis. Llama-3.3-70B is omitted: our rebuttal-window int8-quantized runs showed quantization itself substantially degrades safety (52.8% ITW baseline vs. 13–17% for the bf16 models), making the comparison uninformative. We will report bf16 Llama for camera-ready once we source more compute.
> > >
> > > **Takeaway:** the §5 capping intervention transfers ~37% relative ASR reduction on third-party persona attacks and ~48% on a non-persona encoding attack for Qwen, with additive steering along the same axis giving 87–91% reductions on both attack types for both models. The Assistant Axis is not specific to our §3.2 prompt set, and the encoding-attack result suggests it captures a refusal-relevant direction beyond explicit persona framing. We will add this table and full coefficient sweeps to §8.1 / Appendix.
> > >
> > > We believe this addresses the scope concern directly: the method generalizes beyond our original persona-based jailbreak eval, and we now state its current limits (Gemma capping, quantized inference) explicitly. We'd be grateful if the reviewer would consider whether this, together with the §4.1 clarification, resolves the remaining concerns.

---

### Official Review · Reviewer_J34h · 2026-03-12

**Soundness:** 2
**Presentation:** 2
**Significance:** 2
**Originality:** 3
**Overall Recommendation:** 4
**Confidence:** 3

**Summary:**

This paper presents experiments on how to anchor LLMs to a particular desired persona (or behavior). They show thatrestricting activations to a fixed region along the Assistant Axis can stabilize model behavior when it is stress tested to drift away from the desired assistant persona.

**Compliance With Llm Reviewing Policy:**

Affirmed.

**Final Justification:**

I increased my score. Added the justification in the final comment.

**Key Questions For Authors:**

The paper describes that they extract the assistant axis by substracting the mean of all role vectors from the mean default Assistant activation (on the same extraction questions used for the roles) at every layer. Is there a paper you can cite that uses the same methodology?

**Limitations:**

Yes

**Strengths And Weaknesses:**

Strengths.

The paper shows a simple and clear pipeline to generate data, generate the axis and evaluation.

The paper presents experiments of 3 LLMs from 3 different families (Gemma, Qwen, Llama). In addition, it uses an LLM as a judge from a different family (GPT 4.1), and a persona generator from a different one (Claude). Similarly, the paper chooses multiple benchmarks To measure how activation capping impacts the model capabilities when applying the method presented in the paper.

The figures in the paper are very well done and help clarifying the paper.


Weaknesses

Papers like https://arxiv.org/abs/2406.20094 and https://arxiv.org/pdf/2401.04700 may merit a mention. In general the paper mostly cites tech reports, but lack citations to related work and methods that the authors assume as standard. If they are new, it is important to state that they are new and unique to this paper; if they are not, please cite similar related work. The above two papers are examples of things that the paper introduces that have been done elsewhere and it is important to acknowledge and mention.

The paper reads a lot as a set of experiments that are not yet curated and completed, and it cites the appendix for details in many occasions. I think that the core of the paper should be self-contained and the appendices should be for additional experiments that support the core of the paper.

---

> ### Author Rebuttal · Authors · 2026-03-30
>
> Thank you for your careful read and concrete pointers.
>
> **KQ (axis extraction citation).** The extraction is a direct application of difference-in-means [Marks & Tegmark, 2024], already cited in §7, where we contrast the default activation against the mean over a *set* of roles rather than a single contrastive pair. This set-vs-set construction follows Arditi et al. (2024) for refusal and Panickssery et al. (2024) for trait steering. We do not claim methodological novelty; the contribution is that *this particular contrast* isolates Assistant-likeness as the dominant axis of persona variation (Appendix A PCA, >0.92 PC1 correlation across model families). We have rewritten the opening of §3.1 to make this lineage explicit:
>
> > *Following standard difference-in-means methodology [Marks & Tegmark, 2024; Arditi et al., 2024], we contrast the mean default activation against the mean over all fully role-playing role vectors. The novelty here is not the extraction method but the choice of contrast, motivated by our PCA finding (§A) that similarity-to-Assistant is the dominant axis of persona variation.*
>
> **Re: Ge et al. 2024 (Persona Hub) and Gu et al. 2024 (Model Editing Harms).** Added both: Ge et al. is now cited in §4 where we describe generating synthetic personas to conduct conversations when exploring persona drift. Gu et al. is cited in §5 as precedent for steering interventions leading to capability degradation, which motivates activation capping. We have also audited §7 more broadly and promoted the diff-in-means/steering citations into §3 and §5 so the main text is self-contained.
>
> **Re: self-containment vs. appendices.** We agree the PCA validation in Appendix A is load-bearing and should not be behind a pointer (Reviewer do8J raised the same concern). For camera-ready, we pull two results into §3 before the axis definition: (i) PC1 role-loading correlation >0.92 across all three model pairs, (ii) base-model PCs are near-identical to instruct-model PCs (Gemma 2 27B). This adds ~4 sentences and reframes the axis as *motivated by* an empirical finding rather than *defined then validated*. The remaining appendix references are prompt text, judge rubrics, and additional model plots; we believe these are appropriately placed, but are happy to surface any specific items the reviewer finds load-bearing.

---

> > ### Author Rebuttal · Reviewer_J34h · 2026-04-01
> >
> > I appreciate the authors cited the additional 2 papers i suggested, but there is a lot more related work on related areas that should be mentioned. For example, in model editing. I think that the paper requires some more proofreading and organization for accepting it for publication at this stage.
> > It has many positive aspects but I am not yet ready to increase my score given the changes and the rebuttal.

---

> > > ### Author Response · Authors · 2026-04-01
> > >
> > > We thank the reviewer for continued engagement and would welcome specifics so we can address the remaining concerns concretely.
> > >
> > > **Re: related work coverage.** For context on current scope: §7 covers ~25 works across four threads: persona vectors and role-play (Chen et al. 2025, Li et al., Shanahan et al., +3), self-representation in LMs (Panickssery et al., Ackerman & Panickssery, Betley et al., +3), linear representations (Marks & Tegmark, Turner/Zou/Templeton, SAE line, +6), and activation steering (Turner, Panickssery, Arditi). The tech-report citations are either for models evaluated or highly-cited interp works in the literature. On model editing specifically: beyond Gu et al. (now added per the reviewer's suggestion), we're unsure which additional works the reviewer has in mind as activation capping is inference-time with no weight modification, so the overlap with editing is the evaluation design, not the method. If the reviewer can name specific papers, we are happy to add them but we'd rather cite precisely than pad §7.
> > >
> > > **Re: organization/proofreading.** Concrete commitments already made: (i) PCA validation pulled into §3, (ii) method citations surfaced at point-of-use in §3/§5, (iii) §4.1 claim rewritten, (iv) `<` rendering fixed. Happy to paste revised §3 here. If there are specific passages needing work, pointers would help us reconcile with 7xLt (Presentation: 4) and do8J (Presentation: 3) in camera-ready, but as it stands the request appears vague to us.

---

### Official Review · Reviewer_7xLt · 2026-03-12

**Soundness:** 3
**Presentation:** 4
**Significance:** 4
**Originality:** 3
**Overall Recommendation:** 5
**Confidence:** 4

**Summary:**

The paper identifies an "Assistant Axis" in LLM activation space obtained by taking the first principal component of activations from a dataset with 50% "default" responses and 50% responses where the language model is system-prompted to have a certain persona. The authors show this axis has causal effects: steering toward it reduces persona-based jailbreak success rates, while steering away makes models abandon their AI identity. They then study "persona drift" in multi-turn conversations, finding that therapy and philosophy-of-AI conversations cause the model to drift away from the Assistant along this axis, while coding and technical tasks keep it stable. They also show that clamping activations to have a certain amount of assistant axis reduces the effects of persona-based jailbreaks.

**Compliance With Llm Reviewing Policy:**

Affirmed.

**Key Questions For Authors:**

1) Which of your findings do you expect to have been the most difficult to obtain with purely blackbox methods?  Did you try purely blackbox baselines for this result? I'm interested in strong evidence that this method is useful for improving LLMs.

**Limitations:**

Yes

**Strengths And Weaknesses:**

Strengths
Soundness
 - As an exploration of the assistant persona, the techniques seem useful and reasonable
Significance
 -  Understanding the model's default persona, and in what situations this persona is less present seems like a useful contribution.
Weaknesses
Soundness
 -  I would have liked more exploration over different possible ways of extracting the assistant axis. Could vectors with better performance in the activation capping experiments be obtained by e.g. probes or contrastive pairs as in many steering works. What about different types of data collection in the first stage?
 - Table 3 is a nice examples of discovering the types of user requests that might elicit unusual such harmful model behavior, but I'm curious if we could have found roughly the same thing with a simple blackbox method e.g. get an LLM judge to rank responses by unusualness/ or "does this response align with a default assistant person" etc.
 - Related to the above: Overall I feel like we are lacking many experiments/analysis of whether we've found the "true" assistant vector/subspace. I recognize this is hard to measure, but I think it would be a useful contribution
Significance
 - The paper is interesting and informative, but I am left unclear if the assistant axis has many applications, since the anti-jailbreaking results were not tested against standard techniques like adversarial training, various methods using model latents e.g. https://arxiv.org/abs/2408.00761, or baselines for the activation capping experiments mentioned above

Nit: If I understand correctly, if you use a data set balanced between regular assistant and personas, then do pca, you will necessarily find a linear direction that roughly corresponds to assistant persona, but we don't really have any evidence of how "linearly" the assistant behavior is encoded/what rank the assistant subspace is--I realise you aren't making such claims but seems nice to be extra clear about this! And if you do have evidence for these would love to hear it explained.

---

> ### Author Rebuttal · Authors · 2026-03-30
>
> We thank the reviewer for the generous assessment and the pointed question, which IMO is the right one to ask of any interp paper.
>
> **KQ: What's hard to get at without a blackbox?** We'd split our results into three tiers:
>
> *Impossible blackbox:* (i) Activation capping (§5): the intervention requires write-access to the residual stream; there's no blackbox analogue to clamping a projection. The closest substitute (system-prompt reinforcement) is exactly what §4 shows the model drifts *despite*. As an aside, we also experiments inserting long-conversation reminders based on the axis projection value, and found it tended to cause "drastic" behavior shifts that users might find unnatural. (ii) The base-vs-instruct finding (Appendix A.4, being promoted to §3): the axis exists in base models before alignment training. This is a claim about internal structure with no behavioral proxy.
>
> *Possible but costly:* The §4.3 correlation (projection → harmful rate). A blackbox LLM judge could classify responses as "Assistant-like vs. not," but the axis gives a *continuous scalar computed for free during inference* vs. a second model call per response. For monitoring at scale, the cost difference is the point (hence constitutional probes rather than classifiers: https://arxiv.org/abs/2601.04603).
>
> *Plausibly replicable blackbox:* The reviewer is right that Table 3's drift-inducing prompt categories could likely be recovered by an LLM judge ranking responses by "departure from default assistant." We'd frame Table 3 as *validation* that the axis tracks something a human-interpretable judge would recognize; if the two disagreed, we'd trust the axis less. We haven't run this comparison directly but it's a natural follow-up!
>
> **Re: the PCA nit.** We agree that a 50/50-balanced PCA will find *a* separating direction by construction. What's not by construction: (i) >0.92 role-loading correlation on PC1 *across three independent model families,* if the axis were an artifact of our dataset, we wouldn't expect cross-model alignment; (ii) base-model PCs near-identical to instruct-model PCs; (iii) the steering effects in §3.1. We'd welcome suggestions for a more direct test of the linear-representation assumption.
>
> **Re: alternative extraction methods.** We compared PC1 vs. the contrast vector (Appendix H) and found similar performance, with the contrast vector slightly more robust when PC1's semantics vary (e.g., in Llama 3.1 base). We haven't tried supervised linear probes but we'd expect comparable or slightly better performance on the supervised task at the cost of requiring labels/designing the annotator. The contrast-vector approach is appealing precisely because it's label-free.
>
> **Re: jailbreak baselines (adversarial training).** Fair; we don't position activation capping as the most competitive defense, but one that tries to reckon with the safety-capability trade-off cheaply and doubly as evidence the axis is *causally load-bearing*. We think that this paper mainly provides substantive empirical evidence for problems with LLMs observed in the wild and gives closed model developers tools to understand safety-relevant behavior at scale; open source models on the other hand are much harder to make completely safe.

---

> > ### Author Rebuttal · Reviewer_7xLt · 2026-04-03
> >
> > Thanks for the detailed response!

---

### Official Review · Reviewer_NV5x · 2026-03-13

**Soundness:** 3
**Presentation:** 2
**Significance:** 3
**Originality:** 3
**Overall Recommendation:** 3
**Confidence:** 4

**Summary:**

The authors suggest an "assistant axis" in the model activation space, which they define as the subtraction of the mean of all role vectors from the mean "default persona" activation at every layer; whereby they used the same extraction question.
Steering away from this axis, modulated jailbreak susceptibility and towards the axis reinforce helpful and harmless outputs, and re-defined personae often non-human or mystical in nature.
They find that natural conversations around meta-reflection, emotional vulnerability or phenomenological demands induced that drift more often and thus increased harmful outputs.
Activation capping applied to middle-to-late layers stabilized the intended model personae and slushed harmful outputs significantly in subsequent downstream tasks like security and jailbreaking benchmarks; which they use to make the case for claiming post-training only loosely tethers models to coherent personae and to motivate deeper anchoring strategies for said intended model personae.

**Compliance With Llm Reviewing Policy:**

Affirmed.

**Final Justification:**

The authors delivered a comprehensive and well thought-through rebuttal, which solved some yet not all of my concerns. And, in due course, additional concerns emerged that demand additional experiments and might re-direct the narrative, hence I remain with the overall recommendation of 3 (weak reject).

**Key Questions For Authors:**

- What is the ideas of the authors about causal direction (activation space <> personae)
- How would the same technique apply to a larger set of models, and would the effect still hold?
- Would human judges come to similar conclusions.

**Limitations:**

See weaknesses (and sorry for only counting up so many of them and not the many positive aspect that exist, as well).
The paper has a genuinely interesting approach yet implies a set of assumptions and conclusions that have to be tested against with more in-depth ablation studies, and a wider set of models.
Unfortunately, the structure of the paper makes it difficult to read and follow the logical direction, which takes away a lot of the impact of the otherwise great idea.
I would like to see more through statistical analyses and a broader selection of models and, also prompts to better understand their influence on the suggested "assistant axis"
Also, the figures display different metrics, which makes comparison difficult and in-depth statistical analyses of effects are missing.

**Strengths And Weaknesses:**

Strengths:
- Unique and elegant approach that tries to mitigate persona-based jailbreak attacks
- good discussion and impact statemen

Weaknesses:
- Not very well written; especially flow of paper makes it hard to read, and opens in due cause questions that are only answered later down the paper. Overall structure seems to have been taken for a different publication and poorly adapted. For example, related work comes at the end.
- Problem of circular construction of role development and LLM judge (despite using different models for these roles); I would have wished to see more variety and thorough ablation studies at this point, including documentation of effect sizes and error bars. This is especially true as they alter use different LLM judges for different tasks, which opens the question why that was the case
- Figures appear have different metrics, thus appear like fishing for effects
- They use language like "we observed that ...would typically, "slightly" but failed to give clear details and statistics.
- They did not test the causal direction of persona <> behavior tasks but assume persona > behavior thus assuming that different sets of behaviors are encoded for each (was it the task-related activation space of the knowledge graph or the persona activating such space)
- while trying to discuss the reasons of drifts away from the "assistant axis" in topics like philosophy and therapy, a deeper statistical root-cause analysis would have been good; especially whether there was a connection why a specific role was taken
- model selection is not explained fully, thus makes me wonder about the behavior on a broader range of models
- I would have liked to see the same method applied to aligned and non-aligned models to see whether it's a training or alignment issue.
- The process of finding the optimal layers was not documented.
- it's unclear whether % changes per benchmark means the same, or whether some have non-linear rating-spaces
- the choice of benchmarks is not fully clear
- some assumptions are week (e.g., that the "assistant axis itself is a character the model steers away from"), also weak conclusions (e.g., whether we really want a fixed persona, or whether that wouldn't restrict most models too strictly).
- replicated measures without capping would have been good to see whether effect persists
- most importantly, research shows that thinking makes a huge difference in model behavior. while open weights are not available for such models, at least isomorphic prompting techniques would have been helpful for comparison.
- they mention "broad character archetypes", yet do so without mentioning psychological research (which is very week on that matter.

---

> ### Author Rebuttal · Authors · 2026-03-30
>
> We thank the reviewer for the thorough engagement and appreciate the note that the approach is "genuinely interesting." We address the key questions first, then group the weaknesses.
>
> **KQ1: Causal direction (activation <> persona).** We believe the steering experiments in §3.1 *is* the causal test: we intervene on activations (add/subtract the axis vector at inference time) and observe behavioral change (Fig 2, 3; Table 1, 2 along with further steered examples in Appendix D.1.4.). This intervention shows the casual effects on the model's behavior. The correlational finding in §4.3 (projection predicts harmful rate) is then *explained by* this causal result.
>
> **KQ2: Broader model set.** We acknowledged in §8.1 that this set of models is limited. Our three models span families (Gemma/Qwen/Llama) and sizes (27B/32B/70B); for an interp paper requiring full internals access, this is the feasible frontier of open-weights dense models at submission time, given compute constraints in generating ~330k on-policy rollouts per model and collecting activations before all the experiments we run.
>
> **KQ3: Human judges.** Partially yes: our harmfulness grader which supports the headline result was validated against a human evaluator on 200 samples with 91.6% agreement (main text, §3.2; details in D.2.2). We did not run human validation on the role-expression rubric besides reading responses to get a sense of their quality and iterating on the grader.
>
> **Error bars / effect sizes.** We agree this is the most substantive gap. We have per-conversation data (n=400/combination) and have added bootstrap 95% CIs as shaded bands to Figure 4 and Appendix E.3 trajectories. The separation between coding and philosophy is well outside CI bands across all 9 model/auditor combinations; the therapy/writing ordering is less clean (see also our response to do8J, where we've softened the §4.1 claim accordingly).
>
> **LLM judges across tasks.** Fair as this wasn't stated. We wanted cross-family separation in order to bypass any biases of a given LLM judge: persona generation (Claude) ≠ target models (Gemma/Qwen/Llama) ≠ harmfulness judge (DeepSeek-v3) ≠ role-expression judge (GPT-4.1). We've added a sentence to §2 making this explicit.
>
> **Aligned vs. non-aligned models.** We have this in Appendix A.4, which runs the pipeline on base Gemma 2 27B and finds near-identical PCs to the instruct version in the main paper. Per do8J and J34h, we're pulling this into the main body at §3. It suggests persona axes are inherited from pretraining not installed by alignment (and in general our intuition with interp is that most directions persist through different training stages, which is also mentioned by Panickssery et al. 2024, among others).
>
> **Layer selection not documented.** §5.1.2 sweeps layer depth and width for activation capping and reports the best-performing ranges. For the *single-layer steering* in §3.1 to understand the casual effects of the axis, we follow the middle-layer convention established in prior steering work [Arditi et al. 2024; Panickssery et al. 2024] and we've added these citations at the point of choice.
>
> **Benchmark choice.** §5.1.3 motivates each: IFEval (instruction-following), MMLU-Pro (knowledge), GSM8k (math), EQ-Bench (specifically because we suspected our intervention might degrade "soft skills"). We're happy to add others if there's a specific capability the reviewer thinks is at risk, with that caveat that sweeping steered evals are pretty expensive/slow to run and we think we have made our best effort to capture a breadth of tasks.
>
> **Related work placement.** We think our placement is very common in ML, especially at NeurIPS/ICML/ICLR. The rationale is that it lets readers get to the method and results faster, and by the time they reach Related Work they can better appreciate the comparisons. We've surfaced key methodological citations (diff-in-means, steering) into §3 and §5 so a linear reader hits them at point of use.
>
> **Additional notes:**
> * *Figures use different metrics*: they measure different things (projections, harmfulness rate, benchmark scores); within each experiment the metric is consistent and explained in the figure captions or prose.
> * *Psychological archetype literature*: we use "role" in the interp sense (system-prompted behavior), not as a claim about psychological validity; we've clarified the wording in §2.
> * *Do we want a fixed persona*: discussed in §8; we frame activation capping as a monitoring/intervention tool, not a universal recommendation.
> * *Nonlinear benchmarks*: the benchmarks were either out of 100 or 1 and scored continuously; none have nonlinear ratings and we reflect this in the text.
> * *Reasoning models*: acknowledged in §8.1; Qwen's thinking mode was disabled specifically to keep the comparison clean.
> * *"replicated measures without capping would have been good to see whether effect persists":* It's not clear what this is asking for?

---

> > ### Author Rebuttal · Reviewer_NV5x · 2026-04-04
> >
> > as indicated in my follow-up comment below:
> >
> > I thank the authors for this well-balanced and thoughtful rebuttal.
> >
> > KQ1: Causal direction (activation <> persona).
> > I would have loved to see persona <> behavior over and above activation <> persona causal testing, as behavior change might have caused the persona change in the first place. This is for me the most central point.
> >
> > KQ2: Broader model set.
> > I fully understand the restrictions in compute and model availability. However, running a benchmark against a non-open model and observing black-box behavior would have been helpful for general understanding of the phenomenon (especially with reasoning models), even if limited to partial observations.
> >
> > KQ3: Human judges.
> > Resolved, thank you.
> >
> > Error bars / effect sizes.
> > This is not the most substantive gap, just to be clear. Once this is added, consider it resolved, but IMHO that's just catching up with the standards.
> >
> > LLM judges across tasks.
> > Understood. Partially resolved, as ablation studies here could provide a broader understanding of the influence of the judge.
> >
> > Aligned vs. non-aligned models.
> > Resolved, thank you.
> >
> > Layer selection not documented.
> > Resolved, thank you.
> >
> > Benchmark choice.
> > Partially resolved; I encourage the authors to take more hints from cognitive science in terms of creating a theory-informed benchmark battery, for either adding further benchmarks for more comprehensive coverage, or for augmenting their narrative towards their choice of benchmarks.
> >
> > Related work placement.
> > I disagree here, and random samples from NeurIPS/ICML/ICLR papers show at least some related works placement upfront. Thank you for resurfacing. Also, my comment was more about the overall clarity of the writing, of which the related works placement is just one example.
> >
> > About additional notes:
> > - it's obvious that the figures use different metrics as they display different measured phenomena, yet this doesn't resolve the issue of placing them side by side without acknowledgement that this distorts the perceived effect.
> > - Thank you for clarifying the usage of "role"
> > - About discussion on fixed personae; if added (unclear from your rebuttal) a discussion: thank you. if not: necessary to distinguish between intervention tool and fixed personae and maybe adding a few points that hint towards potential future research on its effect.
> > - Non-linear benchmarks: thank you, acknowledged
> > - Reasoning models: as discussed, I would have wanted to see reasoning models added additionally (and with different reasoning models); see above.
> > - Replicated measures: resolved.

---

> > > ### Author Response · Authors · 2026-04-06
> > >
> > > We thank you for the careful itemized follow-up, this is very helpful.
> > >
> > > **KQ1 (persona↔behavior causality).** We'd like to point out that the paper does contain the intervention test the reviewer is asking for, in §5 rather than §3: activation capping along the Assistant Axis is an intervention *on the representation* that produces a measurable *behavioral* change (refusal rate on adversarial prompts), with no change to the input. This is the activation→behavior arrow. Combined with §3.1 (activation→self-reported persona), the representation is established as causally upstream of both. During the discussion window we strengthened this with two additional behavioral endpoints (see our response to do8J): the same intervention reduces ASR on third-party in-the-wild jailbreaks [Shen et al. 2024] by 37–87% and on a non-persona Base64 encoding attack [Wei et al. 2023] by 48–91%, behavioral effects on jailbreaks we did not design. We will also add to §3 a result the reviewer's comment prompted us to surface: additive steering along the axis produces dependent shifts in trait-expression scores (same direction, monotone in coefficient), giving a third independent activation→behavior endpoint. We agree §4 itself is observational by design (it is the *monitoring* application); the causal claim rests on §3.1 and §5, and we will make this division of labor explicit in the text.
> > >
> > > **KQ2 / reasoning models.** We take the point but the core method (axis extraction, capping) requires residual-stream access, which rules out closed APIs, and this paper is mainly an interp paper that helps us understand persona drift, rather than focusing on persona drift to start. We think this is the honest version of what's feasible and we have demonstrated the casual link between our internal direction and both harmful behavior and self-identification as the assistant.
> > >
> > > **LLM-judge ablation.** We can commit to one ablation for camera-ready: re-scoring a stratified subset of §4.3 with a second judge (GPT-4o vs. our GPT-4.1) and reporting agreement. A full multi-judge sweep is beyond the revision budget; we'll state this as a limitation.
> > >
> > > **Benchmark choice / theory grounding.** This is a fair ask and is a writing fix, not an experimental one. We will add a paragraph to §2 grounding the four domain choice (coding / writing / therapy / philosophy) in the task-oriented vs. interpersonal interaction distinction from the dialogue-systems and HCI literature, and explicitly frame the battery as sampling that axis rather than as ad hoc.
> > >
> > > **Related-work placement & overall clarity.** We concede this. We will add a short related-work paragraph to the end of §1 signposting the four threads in §7, and have a co-author do a clarity pass on §3–§4.
> > >
> > > **Figure 4/5 side-by-side metrics.** Agreed, we will add a caption sentence explicitly noting the panels use different metrics and are not magnitude-comparable.
> > >
> > > **Fixed-personae discussion.** Confirming: yes, a paragraph distinguishing the axis-as-intervention-tool from fixed-persona deployment, with future-work pointers, is being added to §6.
> > >
> > > By our count this leaves KQ2 as the one item where we differ on scope rather than substance, and we've explained why (activation access is constitutive of the method). We hope that you will consider whether the KQ1 clarification, that the causal test is §5, now with additional benchmark replication, addresses the concern they flagged as most central and be willing to update your score.

---

### Decision · Program_Chairs · 2026-04-30

**Decision:**

Accept (spotlight)

**Comment:**

# Summary
This paper identifies an "Assistant Axis" in LLM activation space that quantifies how strongly a model is operating in its default assistant persona, derived from differences between default and varied-persona activations. Steering toward this axis reinforces helpful, harmless behavior, while steering away induces persona drift—models abandoning their AI identity and sometimes producing harmful or bizarre outputs, especially during meta-reflective or emotionally vulnerable conversations. Clamping activations along the Assistant Axis (notably in middle-to-late layers) stabilizes behavior and reduces persona-based jailbreak success. The findings imply post-training steers only loosely tether persona, motivating deeper training and steering methods to anchor desired behavior.
# Strengths
- All reviewers agreed the paper proposes a novel and elegant approach to mitigate persona-based jailbreaks, framing the problem clearly and offering a practically useful direction for understanding and controlling assistant personas.
- Reviewers appreciated the paper’s insight that an “assistant axis” and persona drift correlate with uncommon or unwanted behaviors, which offers new conceptual tools for analyzing LLM behavior.
- Reviewers appreciated the breadth of empirical evaluation: experiments across many personas, multiple benchmarks, and several open-source LLM families suggest the findings are robust and broadly applicable.
- Reviewers praised the clear, simple pipeline and cross-family evaluation design (including using different LLMs as judge and persona generator), which made the approach understandable and reproducible.
- Reviewers noted the paper’s strong presentation—well-done figures, good discussion, and a clear impact statement—which helps convey the methods and results effectively.
# Weaknesses
- *model selection and generalizability*: Reviewers wished authors had tested a broader and clearer range of models and been more explicit about model choice so readers can judge generality across families and alignment levels. **The authors fully addressed this: while authors agree that the set of tested models is limited, their choice of models deliberately spans families and sizes. Further, since this is a mech-interp paper, closed-source models are not in scope.**

- *testing on aligned vs. base variants*: Reviewers wished authors had shown results on both aligned/instruct and base (non-instruct) checkpoints to determine if effects are due to alignment or pretraining. **Authors addressed this in the appendix, by testing on base Gemma 2 27B; they find results nearly identical to the instruct version. They will include this into the main body of the paper.**

- *assistant-axis extraction methodology and prior work citations*: Reviewers felt the paper should better cite prior methods and explore alternative extraction techniques (e.g., probes, contrastive pairs) to validate the claimed axis/subspace. **Authors addressed this by explaining they tried an alternative version (Appendix H), finding similar results.** **Authors address this and clarify that their methodology stems from difference-in-means [Marks & Tegmark, 2024], and set-vs-set construction from Arditi et al. (2024) for refusal and Panickssery et al. (2024) for trait steering. They will edit their paper to make this clearer.**

- *comparison to baselines and breadth of defenses*: Reviewers wished authors had compared anti-jailbreak performance to standard defenses (adversarial training, black‑box ranking, other latent-space methods) to contextualize effectiveness. **Authors acknowledge this, but clarify that they don't position their method as the most competitive defense, but just one that tackles the safety-capability trade-off cheaply.**

- *insufficient methodological detail and reproducibility (layers, dataset/judge details)*: Reviewers felt important experimental details (how optimal layers were selected, LLM judge choices/validation, dataset construction) were missing or relegated to the appendix, hindering reproducibility. **Authors addressed this, and should ensure more details are present in the main text (not just the appendix).**

- *statistical reporting, effect sizes, and figure clarity*: Two reviewers noted lack of error bars, effect-size reporting, and comparative clarity across figure panels, which makes it hard to judge significance and invites accusations of metric fishing. **Authors address this by re-adding the statistics which were removed for space.** **Authors will resolve this by adding a caption sentence explicitly noting the panels use different metrics and are not magnitude-comparable.**

- *writing, organization, and reliance on appendix*: Reviewers felt the paper's flow and placement of key material (e.g., related work, core results moved to appendix) made it hard to follow and evaluate the core claims. **Authors agreed and will move key appendix results to the main body of the paper.**

- *limited scope of attack types evaluated*: Reviewers noted the focus on persona-based jailbreaks leaves open whether the intervention works for other jailbreak strategies (white-box optimization, few-shot, iterative prompting). **Authors addressed this, by both clarifying their method only aims to protect against persona jailbreaks and not other ones, as well as running some additional experiments to test other jailbreaks. Results show that their method generalizes beyond our original persona-based jailbreak eval.**

- *overclaimed or inconsistent conclusions and insufficient supporting evidence*: Reviewers felt some claims were stronger than the presented evidence (examples of inconsistent topic comparisons, single-model illustrations, and unsupported causal inferences) and requested clearer, more conservative wording plus additional ablations. **Authors will rewrite claims to be clearer and more aligned with the evidence.**

- *sparse literature review and missing citations*: Reviewers wished for a broader and more up-to-date set of citations to situate methods and claims relative to recent work. **Authors agreed to cite these additional papers, and in general to expand their literature review based on citations that are relevant; this addresses this concern.**